# Interleukin-6: A New Marker of Advanced-Sarcopenic HCC Cirrhotic Patients

**DOI:** 10.3390/cancers15092406

**Published:** 2023-04-22

**Authors:** Andrea Dalbeni, Leonardo Antonio Natola, Marta Garbin, Mirko Zoncapè, Filippo Cattazzo, Anna Mantovani, Antonio Vella, Stefania Canè, Jasmin Kassem, Michele Bevilacqua, Simone Conci, Tommaso Campagnaro, Andrea Ruzzenente, Alessandra Auriemma, Alessandro Drudi, Giovanna Zanoni, Alfredo Guglielmi, Michele Milella, David Sacerdoti

**Affiliations:** 1Internal Medicine Section C and Liver Unit, Department of Medicine, University of Verona, 37129 Verona, Italy; andrea.dalbeni@aovr.veneto.it (A.D.);; 2Immunology Section, Department of Medicine, University of Verona, 37129 Verona, Italy; 3General Surgery Section, Hepato-Biliary Unit, Department of Surgery, University of Verona, 37129 Verona, Italy; 4Medical Oncology Section, Department of Medicine, University of Verona, 37129 Verona, Italy; 5Radiology Section, Department of Diagnostic and Public Health, University of Verona, 37129 Verona, Italy

**Keywords:** hepatocellular carcinoma, sarcopenia, interleukin 6, Child-Pugh score

## Abstract

**Simple Summary:**

The correlation between Child–Pugh class (CP) and hepatocellular carcinoma (HCC) stage, and between HCC stage and sarcopenia is still not clear. Interleukin 6 (IL-6) promotes the growth of the HCC microenvironment and could promote sarcopenia. We investigated whether IL-6 is correlated with HCC stage and could represent a diagnostic marker for sarcopenia. IL-6 appears to be an effective biomarker for the diagnosis of advanced HCC patients. In addition, IL-6 could be considered a marker of cirrhotic HCC-related sarcopenia, suggesting further investigation with BIA- or CT-dedicated software.

**Abstract:**

Hepatocellular carcinoma (HCC) is the major cause of liver-related death worldwide. Interleukin 6 (IL-6) promotes the growth of the HCC microenvironment. The correlation between Child–Pugh (CP) and HCC stage and between HCC stage and sarcopenia is still not clear. Our aim was to investigate whether IL-6 is correlated with HCC stage and could represent a diagnostic marker for sarcopenia. Ninety-three HCC cirrhotic patients in different stages, according to BCLC-2022 (stages A, B, and C), were enrolled. Anthropometric and biochemical parameters, comprehensive of IL-6, were collected. The skeletal muscle index (SMI) was measured using dedicated software on computer tomography (CT) images. IL-6 level was higher in advanced (BCLC C) compared to the early-intermediate (BCLC A-B) stages (21.4 vs. 7.7 pg/mL, *p* < 0.005). On multivariate analysis, IL-6 levels were statistically dependent on the degree of liver disease severity (CP score) and HCC stages (*p* = 0.001 and *p* = 0.044, respectively). Sarcopenic patients presented lower BMI (24.7 ± 5.3 vs. 28.5 ± 7.0), higher PMN/lymphocyte ratio (2.9 ± 2.4 vs. 2.3 ± 1.2) and increased values of log (IL-6) (1.3 ± 0.6 vs. 1.1 ± 0.3). Univariate logistic regression between sarcopenia and log (IL-6) showed a significant odds ratio (OR 14.88, *p* = 0.044) with an AUC of 0.72. IL-6 appears to be an effective biomarker for the diagnosis of advanced cirrhotic HCC. In addition, IL-6 could be considered a marker of cirrhotic HCC-related sarcopenia, suggesting further investigation with BIA- or CT-dedicated software.

## 1. Introduction

Hepatocellular carcinoma (HCC) represents the fifth most frequent cancer in the world and the fourth leading cause of cancer-related death [1]. Liver fibrosis and cirrhosis, as consequences of chronic inflammation stimulated by different etiopathogenetic factors, such as viruses, alcohol, metabolic risk factors and others, represent in more than 90% of cases the background for HCC development [1,2]. According to the main staging algorithm, the new Barcelona Clinic Liver Cancer (BCLC), a specific treatment is proposed for different HCC stages, from surgery to systemic therapy. The BCLC algorithm, considering tumor status (defined by the number and size of neoplastic nodules, vascular or extra-hepatic invasion), liver function through Child–Pugh (CP) score and tumor-related health status considering patients’ Performance Status (PS); assigns treatment allocation to specific prognostic subclasses [3].

Sarcopenia, a progressive loss of skeletal muscle mass associated with a reduction in muscle function and strength [4,5], is one of the most important predictors of complications, such as cirrhotic decompensation, infections, quality of life and mortality, particularly post-transplantation [4,6]. It has been documented that sarcopenia, in addition to cirrhosis, is frequently associated with chronic pathological pictures, such as chronic obstructive pulmonary disease, cardiovascular diseases and cancer, as well as with drug use and aging [4,6]. A total of 48.1% of patients with liver cirrhosis present with sarcopenia, but despite its high frequency, the pathophysiology of sarcopenia remains to date partially unknown [7]. A growing body of literature highlights how pro-inflammatory cytokines, such as interleukin-6 (IL-6) and tumor necrosis factor-α (TNF-α), contribute to the process of protein homeostasis dysregulation and, consequently, to sarcopenia [7]. Nowadays, sarcopenia is detectable with specific computer tomography (CT) software or through the measurement of appendicular lean mass using bioelectrical impedance analysis (BIA).

To our knowledge, there are not any markers for sarcopenia detection, which link with its severity. Pro-inflammatory cytokines, such as interleukin-6 (IL-6) and tumor necrosis factor-α (TNF-α), contribute to the process of protein homeostasis dysregulation and, consequently, to sarcopenia [7]. In fact, in sarcopenic patients, a negative correlation has been shown between increased serum concentrations of IL-6 and TNF-α and reduced muscle mass and strength, whereas a positive correlation has been reported for HCC and IL-6 [7,8]. In particular, IL6 is a cytokine characterized by pleiotropic activity. In fact, it is able to induce the production, at hepatocyte level, of acute phase proteins, such as fibrinogen, hepcidin, C-reactive protein and serum amyloid A [9]. In addition, IL-6 stimulates the acquired immune response, acting on antibody production and differentiation of cytotoxic T cells [10] and promotes the proliferation and differentiation of non-immune cells, such as synovial and dermal fibroblasts and bone marrow platelets [10]. IL-6 can affect neoplastic progression, acting as an autocrine tumor growth factor as well as evading immunological surveillance. Therefore, high levels of IL-6 should reflect the tumor size [11]. Moreover, through a specific transduction pathway, IL-6 is able, after an initial phase of suppression of p53 activity and, therefore, an alteration of apoptosis, to support the growth of aberrant hepatocytes and promote cancer angiogenesis [12].

The aim of our study was to investigate whether high IL-6 levels are correlated with HCC stage and whether it could represent a diagnostic marker for sarcopenia in patients with HCC and cirrhosis.

## 2. Materials and Methods

This is a single-center retrospective study on prospectively collected data, approved by the local Institutional Ethics Committee (Ethics Committee for Clinical Trials of Verona and Rovigo, 2730CESC-VR/RO), in accordance with the Ethical Principles for Medical Research Involving Human Subjects outlined in the 2013 Declaration of Helsinki. Ninety-three consecutive HCC cirrhotic patients were enrolled from January 2020 to June 2021 in the Liver Unit, Oncology Unit and Hepato-biliary Surgery Unit of Azienda Ospedaliera Universitaria Integrata of Verona. The inclusion criteria were signature of the informed consent for data collection, age over eighteen, previous or recent diagnosis of liver cirrhosis including different etiologies (HCV, HBV, alcoholic, metabolic, autoimmune or others) made through liver biopsy, blood tests and imaging and presence of HCC according to BCLC 2022 criteria not previously treated with systemic therapy [3]. The different etiologies were defined considering the main specific European Association for the Study of the Liver (EASL) guidelines. In particular, we considered as alcohol-related chronic liver disease a daily alcohol consumption of 30 g/day, or a weekly consumption 7 drink units in women and 14 drink units in men [13]. Metabolic chronic liver disease was defined by a biopsy-proven non-alcoholic steatohepatitis or with the exclusion of both secondary causes and alcohol consumption criteria [14,15,16]. The exclusion criteria were heart failure, grade III renal failure and pregnancy.

Anthropometric parameters, anamnestic data, blood chemistry tests and IL-6 assays were performed in all the enrolled patients. In addition, an abdominal CT scan with the evaluation of sarcopenia using dedicated software (see paragraph below) was obtained. The number of lesions, size of HCC nodules (>3 cm or >5 cm), neoplastic thrombosis (portal and/or mesenteric and splanchnic) and/or vascular invasion were registered through CT scan analysis. According to the BCLC 2022 update, the early-intermediate stage of HCC (BCLC A-B) was defined by the presence of one- or two-three nodules smaller than 3 cm or multinodular with preserved liver function, while the advanced stage (BCLC C) was defined by the presence of portal invasion or extrahepatic metastases [3].

### 2.1. IL-6 Assay

The “Abcam Human in vitro ELISA (enzyme-linked immunosorbent assay)” kit was used for the IL-6 assay (Boston, MA, USA) [17]. The sensitivity of the test was 7.8 pg/mL. All patients in the study with values below this limit were, therefore, conventionally assigned the value of 7.7 pg/mL. The IL-6 test was performed in all patients prior to any HCC-specific therapy. The Child–Pugh score was calculated with the same blood sample.

### 2.2. Study of Muscle Mass

To quantify the muscle mass of the patients, CT image analysis was performed with the Slice-O-Matic Version 5.0 (Tomo Vision, Magog, QC, Canada) software. Axial sections were acquired at the level of the third lumbar vertebra (L3). Through the software, it is possible to select specifically the skeletal muscle in a semi-automatic way by previously setting the density values provided by the literature [18]. Specifically, the area of the lumbar transverse muscle section (Lumbar Muscle Cross Sectional Area—LMCA) in cm^2^ was obtained by setting the software to density values between −29 and +150 Hounsfield Units, corresponding to skeletal muscle [18]. The images were then saved in the Digital Imaging and Communications in Medicine (DICOM) format [19]. In sarcopenia studies, one of the main parameters used is the skeletal muscle index (SMI), obtained by normalizing the lumbar muscle area for the patient’s height squared [13]. A lumbar skeletal muscle index value lower than 39 cm^2^/m^2^ for women and 50 cm^2^/m^2^ for men, respectively, corresponding to the cut-offs chosen by the EASL, was used for the diagnosis of sarcopenia [13,20].

### 2.3. Statistical Analysis

The analysis of the data obtained was carried out using Jamovi statistical spreadsheet (Version 2.3, Sydney, Australia) and Python programming language (Beaverton, Oregon, US). Categorical variables were expressed with frequency numbers and percentages, while continuous variables are presented as mean ± standard deviation (SD) or median with interquartile range (IQR), based on data distribution. The verification of the non-normality of the variables was carried out using the Shapiro–Wilk test. The non-parametric Mann–Whitney U test was used for variables with non-normal distribution to find statistically significant differences between two independent populations. The Kruskal–Wallis test was used as a non-parametric equivalent to the one-way ANOVA test to identify a statistically significant difference between multiple groups. The chi-square test of independence was used to evaluate a statistical significance present between two categorical variables.

Binary logistic regression (univariate and multivariate) was used to correlate multiple independent variables with a binary dependent variable. Linear regression was used to identify a possible relationship between a continuous dependent variable and one or more independent variables that are also continuous. In all cases, a statistically significant value was considered when *p* < 0.05.

The ROC (receiver operating characteristic) curve and the AUC (area under the curve) were used to quantify the discriminative capacities of a predictor evaluated with logistic regression.

## 3. Results

### 3.1. General Characteristics and IL-6

Ninety-three consecutive HCC cirrhotic patients were enrolled from January 2020 to June 2021. The general characteristics of the enrolled population (mean age 70 ± 15 years; male 83.8%) are described in Table 1.

Regarding the distribution of cirrhosis etiologies, 29.2% was viral etiology, 27.5% alcoholic, 14.2% metabolic, 26.4% presented an overlap among different etiologies and 7.6% presented other etiopathogenesis.

A total of 60.3% of patients were in CP A, 31.3% in CP B and 8.4% in CP C. Sarcopenia, based on the EASL criteria [20] was present in 57.6% of the population with a mean skeletal muscle index (SMI) of 46.5 ± 12.9 cm^2^/m^2^. Of the 93 patients enrolled, 45 were in HCC early-intermediate stage and 48 in HCC advanced stage. No patient included in our cohort belonged to BCLC 0 and BCLC D stages. Dividing the population by considering HCC stage, CP score was statistically different: patients with early-intermediate stage were in 91.1% of cases with CP A patients, 2.2% CP C, while patients with advanced stage were 31.3% in CP A and 14.5% in CP C.

IL-6 level was higher in advanced compared to early-intermediate stages (21.4 pg/mL vs. 7.7 pg/mL, *p* < 0.005; log (IL-6) 1.45 ± 0.5 vs. 0.99 ± 0.2), but we also documented a statistically significant progressive increase in log (IL-6) in CP A/B/C classes (*p*-value < 0.001), with log (IL-6) values of 0.886, 1.12 and 1.53 in CP A, B and C, respectively. In Figure 1, by contour plot, we documented the trend of CP and log (IL-6) related to HCC stage. It can be observed that patients with early-intermediate stage HCC had a lower CP score and reduced log (IL-6) values; in contrast, the curves of patients with advanced HCC tended toward higher CP and log (IL-6) values (*p*-value =< 0.001). The multicollinearity using variable inflation factors (VIF) was equal to 1 considering IL-6 values and HCC stage.

In multivariate linear regression analysis, IL-6 was found to be statistically dependent on the degree of severity of liver disease (CP score) and on the severity of neoplastic disease, with a *p*-value of <0.001 and 0.044, respectively. No statistically significant dependence was demonstrated for age, gender and BMI (Table 2). Using logistic regression and a cut-off of 0.5, we identify an IL-6 value of 8.13 pg/mL as a predictor of advanced HCC.

### 3.2. Sarcopenia Analysis

Subdividing the total population by considering the EASL sarcopenia criteria, there were statistically significant differences in BMI (*p* = 0.004), advanced HCC (*p* < 0.05), log (IL-6) (*p* = 0.031), lymphocytes (*p* = 0.005) and PMNs/lymphocytes ratio (*p* = 0.017) (Table 3). Specifically, sarcopenic patients presented lower BMI (24.7 ± 5.3 vs. 28.5 ± 7.0), higher PMN/lymphocyte ratios (2.9 ± 2.4 vs. 2.3 ± 1.2) and increased values of IL-6 and log (IL-6) (1.3 ± 0.6 vs. 1.1 ± 0.3).

Univariate logistic regression between sarcopenia and log (IL-6) showed a statistically significant odds ratio (OR 14.88, *p* = 0.044) with an AUC of 0.72, a sensibility of 0.72 and a specificity of 0.57 (Figure 2). The multicollinearity using variable inflation factors (VIF) was 1.

The multivariate binomial regression analysis revealed that only log (IL-6) (*p* = 0.05, Estimate 3.69) demonstrates a nearly significant association with sarcopenia, but not the other variables used in the model (CP score, PMNs/lymphocytes ratio, BMI) (Table 4).

Moreover, the multivariate binomial logistic regression between sarcopenia and all the variables in the model (BMI, log (IL-6), PMNs/lymphocytes, HCC stages and CP score) showed an AUC of 0.86, a sensibility of 0.91 and a specificity of 0.64 (Figure 3).

In another model of multivariate linear regression, an inverse significant correlation was present between the skeletal muscle index (expressed as a continuous variable) and log (IL-6) (Estimate = −7.58) and female sex (Estimate = −10.26), but not with BMI, PMNs/lymphocytes ratio, HCC stages and CP score (Table 5).

In Figure 4, we visually documented, by contour plot, the trend of CP score and log (IL-6), according to the presence or absence of sarcopenia. Non-sarcopenic cirrhotic HCC patients had lower IL-6 levels at the same CP score; in contrast, the curves of sarcopenic patients trend towards higher CP scores and IL-6 values. Using Spearman’s test, IL-6 and sarcopenia had a *p*-value = 0.014; CP and sarcopenia has a *p* = 0.057, while IL-6 and CP had a *p* = 0.007.

## 4. Discussion

In this study, we intended to further investigate whether IL-6 levels correlate with HCC stages in cirrhotic patients.

The first finding is that the increase in IL-6 values in HCC patients correlates with cancer stage, confirming the results of Choi et al. [21]. In the study conducted by Xu et al. [22], the IL-6/STAT3 signaling pathway would promote the development and progression of HCC through numerous mechanisms of action: stimulation of liver cancer stem cell (LCSCs) proliferation, inhibition of p53 onco-suppressor gene transcription and promotion of the expression of anti-apoptotic cytokines, such as Bcl-xl and Bcl-2. In addition, IL-6 receptor activation results in increased expression of vascular endothelial growth factor (VEGF), thus promoting HCC neo-angiogenesis and metastasis, through induction of metalloproteases responsible for basement membrane degradation. In the study by Myoiin et al. [23], response to systemic therapy with Atezolizumab/Bevacizumab was observed using CT or MRI in 64 patients with advanced HCC. Thirty-four plasma proteins were measured in the enrolled patients, demonstrating a significant correlation only for IL-6 and INF-alpha with disease progression. It was documented that high IL-6 levels were associated with a higher rate of macrovascular invasion compared with patients with lower plasma values, suggesting not only a positive correlation between IL-6 concentration and neoplastic progression but also a worse prognosis of patients undergoing immunotherapy. In agreement with our results, Myoiin and colleagues described a correlation between elevated plasma levels of IL-6 and advanced stage of disease; however, the patients enrolled were all with CP A; therefore, they were not able to show, as described in our study, a correlation between cirrhosis progression and IL-6 levels. Higher values of IL-6 in Myoiin’s study seem to be closely related to the neoplasm alone, and to the lower responsiveness to therapy, as also described in the studies of Xu et al. and Shao et al. [22,24].

In the study by Xu et al. [22], the role of the IL-6/STAT3 pathway in the progression and recurrence of HCC and the possibility of its inhibition was described as a possible future therapeutic approach, specifically through monoclonal antibodies specific for IL-6 (Siltuximab and ADL518) and IL-6R (Tocilizumab) or STAT3-activated Jak inhibitors (Ruxolitinib) or STAT3 inhibitors (LLL12 andC188-9). In our study, the correlation between high IL-6 levels and advanced HCC stage was concordant. Moreover, the correlation of plasma IL-6 values with the severity of hepatopathy, an aspect not shown in the studies by Shao et al. [24] and Miojin et al. [23], was also demonstrated. For these reasons, IL-6 could be studied as a biomarker of cancer stage.

Another interesting finding is that IL-6 emerges for the first time as a potential specific marker of sarcopenia in HCC cirrhotic patients (VIF = 1; OR 14.8, *p* = 0.04). Moreover, non-sarcopenic patients exhibited lower IL-6 levels at the same CP class, while, in contrast, sarcopenic patients tended to present higher CP class and IL-6 values or other markers of systemic inflammation, such as the PMNs/lymphocyte ratio. The correlation between sarcopenia and IL-6 has already been described for several types of cancer, suggesting that IL-6 is a central regulator of cancer progression and cancer-associated cachexia [25,26,27]. Furthermore, several studies evaluated the effects of IL-6 inhibitors (such as Clazakizumab or Tocilizumab) on increased body weight but only few case reports ameliorated cancer-associated cachexia [28,29,30]. Although sex is a considered variable in the detection of sarcopenia, gender resulted a significant variable in the multivariate analysis. The correlation between SMI and female gender could reflect the possible effect of menopause, as the mean age of our cohort was 70 years old and the high prevalence of sarcopenia in postmenopausal women has already been described [31].

To our knowledge, only Choi et al. [21] described in HCC patients a correlation between three different cytokines (including IL-6), sarcopenia and survival; however, that study did not propose IL-6 as a marker of sarcopenia. From our data, we believe that IL-6 could be useful to detect HCC, especially in case of tumor infiltration, and possibly sarcopenia in all stages of cirrhosis (CP classes A, B or C).

Our study has some limitations: first, this is a single-center study with a relatively small sample. The lack of a cirrhotic control group to clarify the role of IL-6 in predicting sarcopenia in this context alone could also be considered a limitation. Moreover, the lack of a control group consisting of non-cirrhotic HCC patients could be also considered a limitation. IL-6 is a cytokine released during inflammatory processes: it is involved in innate and adaptive immune responses, but also in the activation of metabolic and catabolic pathways in muscle tissue, depending on patients’ condition and muscle state [31]. When there is prolonged exposure to IL-6, muscle catabolism and atrophy are observed, probably due to a lower level of muscle anabolism and activation of different energy production pathways [31].

Hence, IL-6 could be considered a clinical biomarker of sarcopenia and general frailty in patients. In their review, Picca et al. highlighted how, in sarcopenic and frail subjects, there was a chronic state of low-grade inflammation, sustained by various stimuli (i.e., gut microbiota, pathogens, cellular debris or misplaced intracellular components). As a consequence, several cytokines (and especially IL-1, IL-6 and TNF-α) exhibited an important role in this context. In particular, IL-6 was associated with frailty and sarcopenia in people aged 75 years [32,33].

Future studies should focus also on non-HCC cirrhotic patients to confirm our observations and to compare the usefulness of IL-6 in both HCC and non-HCC cirrhotic patients.

IL-6, as demonstrated by the logistic regression and multicollinearity model, could be considered a marker of cirrhotic HCC-related sarcopenia, suggesting further investigation with BIA- or CT-dedicated software.

## 5. Conclusions

In conclusion, our study showed that IL-6 can be considered a useful biomarker for HCC stages, as well as a possible target for future therapies showing a direct role in cancer development. Moreover, IL-6 emerges for the first time as a potential specific marker of sarcopenia in HCC cirrhotic patients, which is useful especially when other diagnostic tools for detecting early stages of sarcopenia are not available.

## Figures and Tables

**Figure 1 cancers-15-02406-f001:**
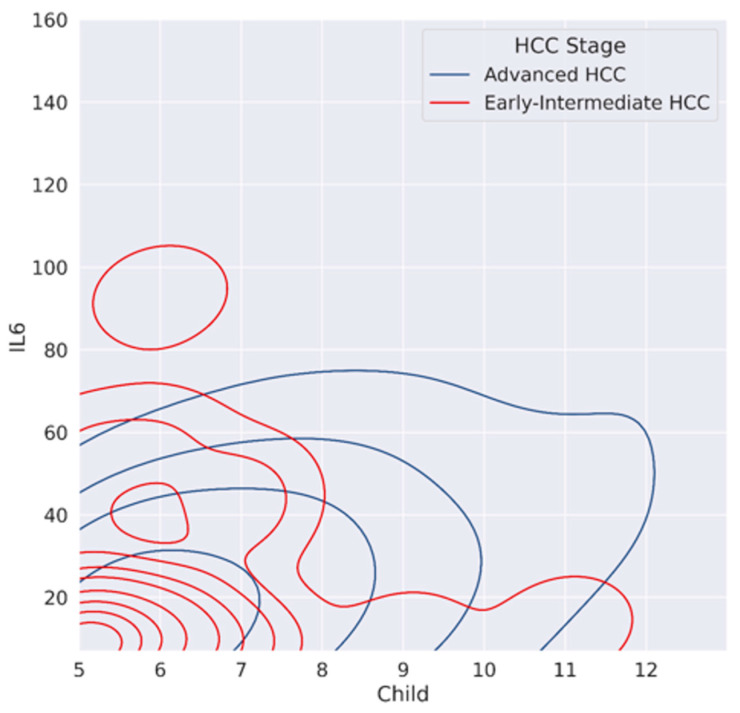
Contour plot between IL 6, Child–Pugh score and HCC stages.

**Figure 2 cancers-15-02406-f002:**
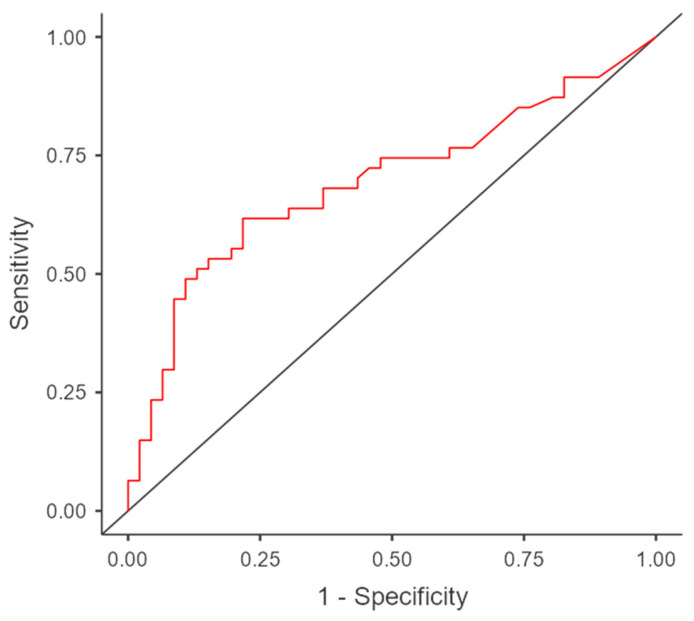
ROC Curve between sarcopenia and log (IL-6).

**Figure 3 cancers-15-02406-f003:**
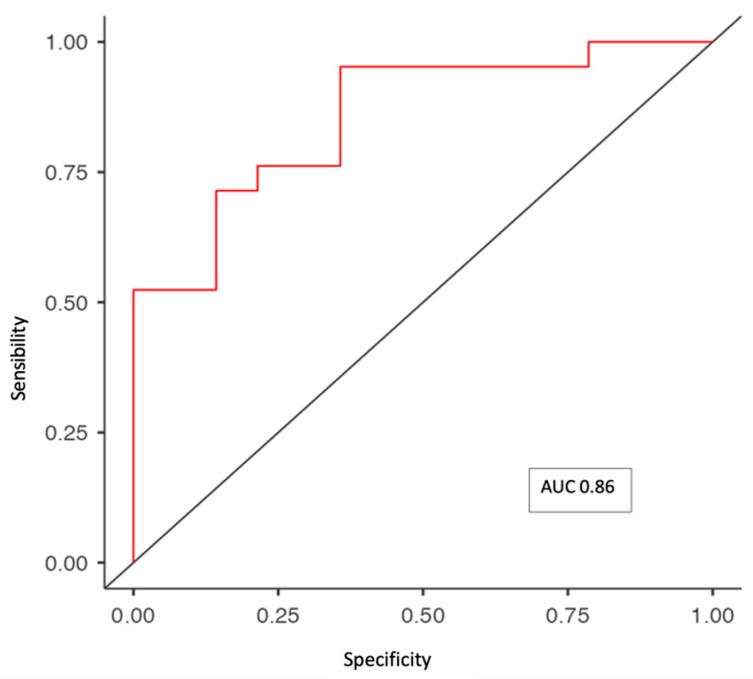
ROC curve between sarcopenia and variables in the multivariate analysis.

**Figure 4 cancers-15-02406-f004:**
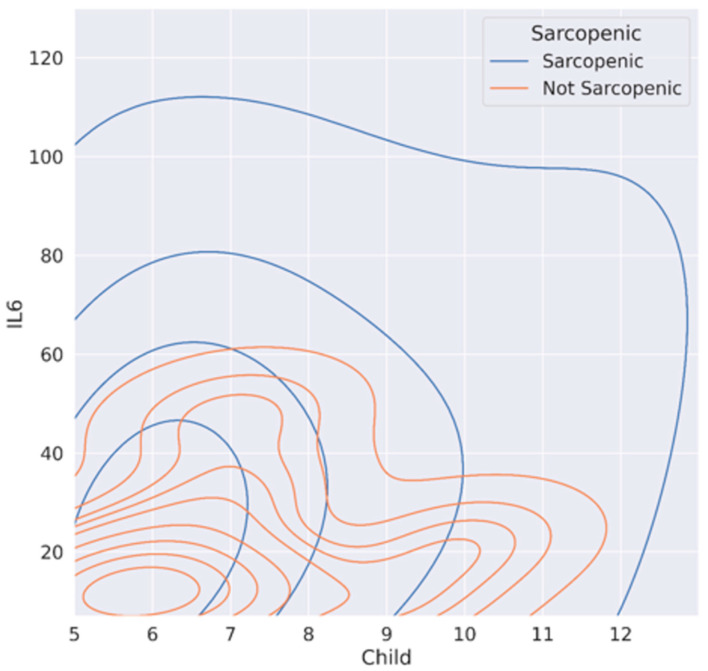
Contour plot between IL-6, Child–Pugh score and sarcopenia.

**Table 1 cancers-15-02406-t001:** General characteristics of the studied population subdivided considering HCC stages. Values are presented as percentages, mean ± SD or median (IQR); *p*-values are presented when statistically significant and highlighted in bold.

Variable	Total Population (n = 93)	Early-Intermediate Stage (BCLC A-B) HCC Patients (n = 45)	Advanced Stage (BCLC C) HCC Patients (n = 48)	*p* Value
Sex (M)	83.8%	86.7%	83.3%	ns
Age (±SD)	70.0 ± 15.0	69.0 ± 17.0	72.0 ± 12.3	ns
BMI kg/m^2^ (±SD)	26.2 ± 6.5	27.2 ± 5.6	26.2 ± 5.9	ns
BMI > 30	37.1%	38.4%	36.0%	ns
**Etiopathogenesis**				ns
Viral	29.2%	28.9%	33.3%	
Alcoholic	27.5%	28.9%	18.7%	
Metabolic	14.2%	13.3%	16.7%	
Overlap	26.4%	26.7%	27.1%	
Other	3.6%	2.2%	4.2%	
**BCLC Stages**				
BCLC 0	0%	-	-	
BCLC A	19.3%	40%	-	<0.001
BCLC B	29%	60%	-	
BCLC C	51.6%	-	100%	
BCLC D	0%	-	-	
**Child-Pugh class**				<0.001
A	60.3%	91.1%	31.3%	
B	31.3%	6.7%	54.2%	
C	8.4%	2.2%	14.5%	
MELD score	9 ± 5.1	9 ± 5.0	9.5 ± 5.3	ns
IL-6 (pg/mL)	7.7 (8.3)	7.7 (0.7)	21.4 (14.4)	0.042
log IL-6	1.1 ± 0.3	0.9 ± 0.2	1.4 ± 0.5	0.042
AFP (kU/L)	5.0 (33.1)	5.5 (10.8)	717 (1272)	0.005
Log AFP	1.3 ± 1.1	1.1 ± 0.9	1.5 ± 2.0	0.005
PMN (10^9^/L)	3.2 ± 2.4	3.7 ± 2.2	2.9 ± 1.7	ns
Lymphocytes (10^9^/L)	1.1 ± 0.7	1.4 ± 0.8	1.0 ± 0.7	ns
PLTs (10^9^/L)	128 ± 85	129 ± 99	121 ± 72.3	ns
WBC (10^9^/L)	5.6 ± 3	5.7 ± 2.4	5.1 ± 2.4	ns
Hb (g/L)	125 ± 30.5	134 ± 28	126 ± 26.3	<0.001
Creatinine (mg/dL)	0.8 ± 0.4	0.8 ± 0.3	0.8 ± 0.4	ns
PMNs/Lymphocytes	2.7 ± 2.2	2.7 ± 1.5	2.8 ± 2.3	ns
Skeletal muscle index (SMI) (cm^2^/m^2^)	46.5 ± 12.9	48.4 ± 16.1	46.0 ± 10.7	ns
Sarcopenia *	59.7%	50.0%	68.7%	0.05

Legend: HCC, hepatocarcinoma; AFP, alfa-fetoprotein; BMI, body mass index; BCLC, Barcelona clinic liver cancer; Hb, hemoglobin; IL-6, interleukin-6; MELD, model for end-stage liver disease; PLTs, platelets; PMNs, polymorphonucleates; WBC, white blood cells; ns, not significant. * not available for one patient.

**Table 2 cancers-15-02406-t002:** Multivariate linear regression considering IL6 and other variables.

	Estimate	t	*p*
Age	0.60	1.54	ns
Gender	−9.31	−1.04	ns
HCC stages	26.39	2.04	0.044
CP score	7.37	3.58	<0.001
BMI	0.85	1.62	ns

Legend: HCC, hepatocellular carcinoma; CP, Child-Pugh; BMI, body mass index; ns, not significant.

**Table 3 cancers-15-02406-t003:** Mann–Whitney U test for sarcopenic and non-sarcopenic HCC cirrhotic patients. Values are presented as percentage, mean ± SD or median (IQR), *p*-values are presented when statistically significant and highlighted in bold.

Variable	Non-Sarcopenic Patients (n = 37)	Sarcopenic Patients (n = 55)	*p* Value
Sex (M)	36.1%	45.8%	ns
Age	68.0 ± 11.8	72.0 ± 15.0	ns
BMI, (kg/m^2^)	28.5 ± 7.0	24.7 ± 5.3	0.004
BMI > 30	50%	28%	ns
Child–Pugh score A	70%	67%	ns
MELD score	8.0 ± 3.0	9.0 ± 4.0	ns
HCC early-intermediate stage	59.5%	40%	ns
HCC advance stage	40.5%	60%	<0.005
IL-6 (pg/mL)	7.7 (8.3)	10.4 (28)	<0.05
log IL-6	1.1 ± 0.3	1.3 ± 0.6	0.031
AFP (kU/L)	7.5 (77)	7.0 (40)	ns
PMNs (×10^9^/L)	2.9 ± 1.8	3.2 ± 2.5	ns
Lymphocytes (×10^9^/L)	1.4 ± 0.9	1.0 ± 0.5	0.005
PLTs (×10^9^/L)	131.0 ± 65.0	128.0 ± 83.0	ns
WBC (×10^9^/L)	5.4 ± 2.5	5.3 ± 3.5	ns
Hb (g/L)	132.0 ± 21.5	130.0 ± 30.0	ns
Creatinine (mg/dL)	0.9 ± 0.3	0.8 ± 0.3	ns
PMNs/Lymphocytes ratio	2.3 ± 1.2	2.9 ± 2.4	0.017

Legend: AFP, alfa-fetoprotein; BMI, body mass index; Hb, hemoglobin; HCC, hepatocellular carcinoma, IL-6, interleukin-6; MELD, model for end-stage liver disease; PLTs, platelets; PMNs, polymorphonucleates; WBC, white blood cells; ns, not significant.

**Table 4 cancers-15-02406-t004:** Univariate and multivariate binomial regression between sarcopenia and other variables.

	Univariate Analysis			Multivariate Analysis		
	Estimate	t	*p*	Estimate	t	*p*
BMI > 30	−0.91	−1.27	0.20	−1.74	−1.85	ns
Log (IL-6)	2.7	2.0	0.044	3.63	1.89	0.05
PMNs/Lymphocytes	0.18	1.42	0.15	0.13	0.79	ns
HCC stages	−0.41	−0.57	0.56	−0.36	−0.41	ns
CP score	−0.38	−0.55	0.58	−0.73	−0.42	ns

Legend: BMI, body mass index; IL-6, interleukin-6; PMNs, polymorphonucleates; HCC, hepatocellular carcinoma; CP, Child-Pugh; ns, not significant.

**Table 5 cancers-15-02406-t005:** Univariate and multivariate linear regression between SMI and other variables.

	Univariate Analysis			Multivariate Analysis		
	Estimate	t	*p*	Estimate	t	*p*
Sex (females)	−7.85	−1.85	0.07	−10.26	11.69	0.02
BMI	0.60	1.94	0.06	−0.90	−0.20	ns
Log (IL-6)	−2.27	−1.43	0.16	−7.58	−2.13	0.04
PMNs/Lymphocytes	−0.39	−1.39	0.17	−0.07	−0.18	ns
HCC stages	0.67	0.23	0.82	−1.20	−0.40	ns
CP score	3.18	1.14	0.26	−0.20	−0.27	ns

Legend: BMI, body mass index; IL-6, interleukin-6; PMNs, polymorphonucleates; HCC, hepatocellular carcinoma; CP, Child-Pugh; ns, not significant.

## Data Availability

All data are explicated within the article and tables. Any other clarification is available on request from the corresponding author.

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
