# Peer review of "Interleukin-6: A New Marker of Advanced-Sarcopenic HCC Cirrhotic Patients"

_cancers, 2023, doi:10.3390/cancers15092406_

Round 1
Reviewer 1 Report (Previous Reviewer 1)
Dalbeni A et al investigated whether IL6 is correlated with HCC stage and it could represent a diagnostic maker for sarcopenia. In order to showed this the cohort (93 cirrhotic HCC patients) was splited the patients into 2 groups early- intermediate stage and advanced stage. After performing a multivariate analysis Il6 was dependent of cirrhosis severity (CP) and HCC stage. In a second step, the authors showed that sarcopenia correlates with IL6 and gender but not with Child Pugh and BCLC stage. The conclusion of the study is that IL6 appears to be biomarker for the diagnosis of advanced cirrhotic HCC patients and it could be a marker for cirrhotic HCC sarcopenic patients
The manuscript has several issues that might conduct to a misleading conclusion.
1. The authors staged patients with Child Pugh C (not preserved liver function) as BCLC A, B or C. BCLC. According to BCLC, criteria patients included in these groups need to have a preserved liver function, which is not the case of Child Pugh C.
Could explain the authors why they consider Child Pugh C as patients with preserved liver function?
2. It is not clear how the authors selected the parameters that they included in the univariate analysis. Thus, it is not clear why the authors did not include in the univariate analysis the sarcopenia when they evaluated the correlation between IL 6 and HCC stage.
Could the authors explain, based on which criteria the parameters included in the univariate analysis in both cases ( Table 4 and Table5) were selected.
3. The high number of patients staged as Child Puh A ( 70%- non sarcopenic , 67 % sarcopenic) might be the cause for the lack of correlation between sarcopenia and Child Pugh, fact that migh lead us to a false conclusion that there is no correlation with te grade of impaired liver function. Moreover, it is not clear how the authors calculated the percentages of HCC advanced stage and early- intermediate because the sum is in the non sarcopenic group below 100% while in the sarcopenic group is above 100%. In looks that they calculated the percentage within the HCC stage group , but this is misleading since we do not know how many patients are staged as Early – intermediate or advanced in sarcopenic and non-sarcopenic group. To this, it is added the fact that CP C are in fact BCLC D.
Could you please clarify?
4. To determine the value of a new biomarker, it is not sufficient to show that it is significantly related to the outcome, statistically significant in a multivariable model including the standard clinical and pathologic factors, or more significant than the standard clinical and pathologic factors. It is known that a parameter that is statistically significant in a multivariable model might not substantively improve the model's predictive accuracy. P-values and odds or hazard ratios do not meaningfully describe a biomarkers' ability to classify patients. For a biomarker to be potentially clinically useful, it is necessary to show that adding the biomarker to an existing model based on the most important clinical and pathologic factors substantively improves the predictive accuracy (discrimination and calibration) of the model. Moreover there is no validation cohort.
Author Response
Please see the attachment.

Reviewer 2 Report (Previous Reviewer 3)
This manuscript is well revised.
Author Response
Thank you for your time and revision, we appreciate that.
Reviewer 3 Report (New Reviewer)
This paper by Dalbeni et al. is clear and relevant to the field. The limitations raised in the Discussion part are very important and have been well written by the authors. I endorse publication of this work.
Author Response
Thank you for your time, revision and comments, we appreciate that.
This manuscript is a resubmission of an earlier submission. The following is a list of the peer review reports and author responses from that submission.
Round 1
Reviewer 1 Report
In the current manuscript, the authors aimed to identify if IL6 correlates with HCC stages and if this biomarker could diagnose sarcopenic HCC patients. A population of 93 HCC patients (sarcopenic and non-sarcopenic) was analysed. The statistical analysis was done in 2 steps: a) IL 6 correlation with HCC stage (early/ advanced), Child-Pugh score, Gender, Age); b) sarcopenia correlation with HCC stage (early/ advanced), Child-Pugh score, gender, age. After performing the statistical analysis, the authors concluded that:1) IL-6 appears to be an effective biomarker for diagnosing advanced HCC, especially in CP A patients; 2) IL-6 could be considered a marker of HCC-related sarcopenia.
Unfortunately, the manuscript has some critical weaknesses.
1)Abstract - Aim: The authors aimed to identify the following:
a) A possible correlation between IL-6 and HCC stage. However, the authors split the HCC cohort only into two groups (early and advanced) and not into five stages (BCLC 0-D), as the BCLC staging system recommends. Moreover, according to the authors, in the current study, the advanced stage comprises the intermediary and an advanced stage according to BCLC staging (Line 101-102). We ask the authors to clarify this point.
b) Il6 could represent a diagnostic marker for sarcopenic-HCC. Using the wording "sarcopenic-HCC", the authors suggest that cirrhosis is not a significant factor, and the results contradict this. We ask the authors to clarify this point.
2) Abstract: Line 29-30: In multivariate analysis, IL-6 was linked with cancer stage, CP, age and gender, while in the results section ( line 188-190). In multivariate linear regression analysis, IL-6, no statistical significance was found with age, sex and BMI.
3) Method section: Were the study patients recruited before any therapy was started? This point is essential since systemic therapy is known to accentuate sarcopenia.
4) Method section: The authors should mention if the study was a prospective or retrospective analysis of a prospective cohort.
5) Result section: In the first table, the following data are not mentioned: a) how many patients were in every BCLC stage (0-D), b) how many patients are cirrhotic/non-cirrhotic?
6) Result section: Figure 2 is not mentioned in the text. Please mention and explain it.
7) Result section: "The multivariate linear regression showed a significant inverse correlation between SMI and log (IL-6) (Estimate= -7.58) and female sex (Estimate= -203 10.26), but not with BMI, lymphocytes, HCC stages and CP score". From the above result, the authors used the SMI as a continuous variable and not as a categorical variable (sarcopenic/non-sarcopenic). Could the authors confirm/infirm that was used the value of SMI was used as a continuous variable? Suppose it was used as a continuous variable. Could the authors explain why they opted for this variant since the analysis was done to show the correlation between sarcopenia and IL6, HCC stage, CP, BMI. Moreover, in this case, the authors identified an inverse correlation between skeletal muscle index and IL6 level and not between sarcopenia and IL6.
8) Discussion paragraph: Based on this study's results, the authors suggest using IL 6 to detect HCC, especially in cases of tumour infiltration, and possibly sarcopenia in all stages of cirrhosis (CP score A, B or C), but especially in CP score A. Here are several points that should be clarified: a) which level of IL6 can be used for diagnosis? b) what do the authors consider advanced HCC? In the current study, the advanced stage included BCLC B, BCLC C, and probably BCLC D, but in the above statement is mentioned, only BCLC C, c) Why do the authors state, especially in Child Pugh A? Did the authors perform a supplementary analysis for every Child-Pugh class? Please clarify. We ask the authors to clarify this aspect or reformulate it.
9) Discussion paragraph: The correlation between sarcopenia and gender is not discussed
10) Discussion paragraph: The lack of correlation between Il6 and age was not discussed
11) Discussion paragraph: In the discussion section, the authors discuss the link between systemic therapy and IL6, while this aspect was not investigated in the current study
12) Conclusion IL6 is a marker of HCC-related sarcopenia. In this form, it is suggested that IL6 is a marker only for HCC sarcopenia, ignoring cirrhosis.
13) Conclusion section: Line 300-301: Moreover, IL-6 emerges for the first time as a potential specific marker of sarcopenia in HCC patients, helpful especially when other diagnostic tools for detecting early stages of sarcopenia are not available. Did the authors evaluate graded sarcopenia (early sarcopenia, advanced sarcopenia)? In the results section, er found no information concerning this aspect.
Author Response
1) Abstract - Aim The authors aimed to identify the following:
a) A possible correlation between IL-6 and HCC stage. However, the authors split the HCC cohort only into two groups (early and advanced) and not into five stages (BCLC 0-D), as the BCLC staging system recommends. Moreover, according to the authors, in the current study, the advanced stage comprises the intermediary and an advanced stage according to BCLC staging (Line 101-102). We ask the authors to clarify this point
Thank you for the opportunity to clarify this point. We have specified both in the abstract and in the Materials and Methods section the definition of early-intermediate (stages A-B) and advanced HCC (stage C). Moreover, no patients included in our cohort belonged to BCLC stage 0 and BCLC stage D.
b) Il6 could represent a diagnostic marker for sarcopenic-HCC. Using the wording "sarcopenic-HCC", the authors suggest that cirrhosis is not a significant factor, and the results contradict this. We ask the authors to clarify this point.
Thank you for the suggestion. The study was conducted only on cirrhotic patients, as is also stated in the Materials and Methods section: “93 consecutive HCC cirrhotic patients were enrolled”.
But, as you correctly pointed out, the “sarcopenic-HCC” definition can be misleading. Therefore, we have corrected this in the abstract and highlighted that the study population consists of patients with cirrhosis and HCC.
c) Abstract: Line 29-30: In multivariate analysis, IL-6 was linked with cancer stage, CP, age and gender, while in the results section ( line 188-190). In multivariate linear regression analysis, IL-6, no statistical significance was found with age, sex and BMI.
Thank you for your comment and sorry for the inaccuracy. In our study, we explored the correlation between IL-6, neoplastic stage, Child-Pugh score, age and gender, and in a multivariate linear regression IL-6 was found to be statistically dependent on the degree of liver disease (CP score) and neoplastic disease severity, with a p value of 0.001 and 0.044, respectively. Instead, no statistically significance correlation was demonstrated with age, gender and BMI. Thus, we corrected the text in the Abstract section as follows:
“On multivariate analysis, IL-6 levels were statistically correlated with severity of liver disease (CP score) and HCC stages (p=0.001, p=0.044, respectively).”
2) Method section: Were the study patients recruited before any therapy was started? This point is essential since systemic therapy is known to accentuate sarcopenia.
Thank you for the comment, we agree. In fact, we have included only patients that were not previously treated with systemic therapy. We specified this point in the Methods section, also regarding the timing for IL-6 assay.
3) Method section: The authors should mention if the study was a prospective or retrospective analysis of a prospective cohort.
Thank you for this suggestion, we specified in the “Materials and Methods” section (line 87) that this is a retrospective study of prospectively collected data.
4) Result section: In the first table, the following data are not mentioned:
- how many patients were in every BCLC stage (0-D),
Thanks for the request. In the result section we specified that in our cohort there were no patients belonging to BCLC stage 0 or D. We also amended in the text the word “early stage” with the more correct “early-intermediate” stage (considering BCLC stage A and B). Moreover, we added in Table 1 the number of patients for each BLCL stage.
2. how many patients are cirrhotic/non-cirrhotic?
Thanks for the question. We clarified in the title and inclusion criteria that all patients were cirrhotic.
5) Result section: Figure 2 is not mentioned in the text. Please mention and explain it.
Figure 2 is described at line 276-280.
6) Result section: "The multivariate linear regression showed a significant inverse correlation between SMI and log (IL-6) (Estimate= -7.58) and female sex (Estimate= -10.26), but not with BMI, lymphocytes, HCC stages and CP score". From the above result, the authors used the SMI as a continuous variable and not as a categorical variable (sarcopenic/non-sarcopenic). Could the authors confirm/infirm that was used the value of SMI was used as a continuous variable? Suppose it was used as a continuous variable.
Thanks for the question; we confirm that SMI is a continuous variable (table 5). We have added also the multivariate analysis using sarcopenia as a categorical variable in the text and in Table 4.
7) Could the authors explain why they opted for this variant since the analysis was done to show the correlation between sarcopenia and IL6, HCC stage, CP, BMI. Moreover, in this case, the authors identified an inverse correlation between skeletal muscle index and IL6 level and not between sarcopenia and IL6.
Thanks for the request. We added more information in the new version of the draft. However, we confirm the inverse correlation between SMI and IL-6, while the correlation between sarcopenia and IL-6 nearly reached significance (p=0.05).
8) Discussion paragraph: Based on this study's results, the authors suggest using IL 6 to detect HCC, especially in cases of tumour infiltration, and possibly sarcopenia in all stages of cirrhosis (CP score A, B or C), but especially in CP score A. Here are several points that should be clarified:
a. which level of IL6 can be used for diagnosis?
Thanks for the suggestion. Using logistic regression and cut off 0,5 we identify a value of 8,13 pg/ml as predictor of advanced HCC. We added this in the text (line 212).
b. what do the authors consider advanced HCC? In the current study, the advanced stage included BCLC B, BCLC C, and probably BCLC D, but in the above statement is mentioned, only BCLC C,
Thanks for the required clarification, we have corrected and added in the text. As we stated above, no patients were in BCLC D stage, we considered as advance stage patients belonging to BCLC C stage.
c. Why do the authors state, especially in Child Pugh A? Did the authors perform a supplementary analysis for every Child-Pugh class? Please clarify. We ask the authors to clarify this aspect or reformulate it.
Thanks for the suggestion. We did not perform a supplementary analysis for every CP class (there is a Contour plot considering CP score in Figure 1). We have reformulated this aspect in the text.
9) Discussion paragraph: The correlation between sarcopenia and gender is not discussed
Thanks for the request. We discussed that aspect in the discussion section (line 339-343)
“Although sex is a considered variable in the detection of sarcopenia, gender resulted a significant variable in the multivariate analysis. The correlation between SMI and female gender could reflect the possible effect of menopause, as the mean age of our cohort was 70 years old and the high prevalence of sarcopenia in postmenopausal women has already been described [31].”
10) Discussion paragraph: The lack of correlation between Il6 and age was not discussed
Thanks for the request. We didn’t discuss in the text because in all the analysis age was not significant.
11) Discussion paragraph: In the discussion section, the authors discuss the link between systemic therapy and IL6, while this aspect was not investigated in the current study
Thanks for the suggestion; we removed this part from the discussion paragraph considering that no patient had started systemic therapy before the IL-6 assay.
12) Conclusion IL6 is a marker of HCC-related sarcopenia. In this form, it is suggested that IL6 is a marker only for HCC sarcopenia, ignoring cirrhosis.
IL-6 levels were shown to progressively increase with Child-Pugh score and, at the same time, to be significantly higher in advanced sarcopenic HCC stage. Since advanced HCC stage patients belonged more frequently to higher CP score, IL-6 could be hypothesized to be related only to the severity of liver disease instead of cancer stage. However, the multicollinearity between variables (HCC stage, sarcopenia and CP score) was found to be lower and thus confirming that, despite IL-6 linearly increase with disease severity, the relationship between cancer stage, IL-6 and sarcopenia is still proven (line 248).
13) Conclusion section: Line 300-301: Moreover, IL-6 emerges for the first time as a potential specific marker of sarcopenia in HCC patients, helpful especially when other diagnostic tools for detecting early stages of sarcopenia are not available. Did the authors evaluate graded sarcopenia (early sarcopenia, advanced sarcopenia)? In the results section, er found no information concerning this aspect.
Sarcopenia, defined according to EASL guidelines (SMI ≤ 39 cm2/m2 for women and 50 cm2/m2 for men, respectively) was not further sub-classified since a clear and standardized classification of early and advanced sarcopenia does not exist. For this reason, SMI was considered as a linear variable in regression analysis.
Reviewer 2 Report
The expression of IL-6 during chronic hepatic inflammation activates downstream targets of STAT3 transcription factor, which drives neoplastic transformation in the hepatic microenvironment. In this article, Dalbeni et al. investigated the relationship between IL-6 and sarcopenic-HCC. They found that IL-6 was an effective biomarker for the diagnosis of advanced HCC. In addition, IL-6 could be considered as a marker of HCC-related sarcopenia. Although their findings are clinically interest, several critical points are worthy of attention.
Specific comments:
1. The limitation is the lack of control group (non-HCC cirrhotic patients). Thus, the relationship between IL6 and sarcopenia needs further clarify.
2. The etiologies of HCC should be defined in more detail. For example, the definition of alcoholic and metabolic HCC should be described in more detail.
3. In the section 3.2 Sarcopenia analysis, the results of multivariate linear regression should be provided as a table.
4. The evaluation of the diagnostic performance of IL6 showed that AUC values were poor (0.7). It maybe not a good diagnostic biomarker for HCC related sarcopenia.
Author Response
- The limitation is the lack of control group (non-HCC cirrhotic patients). Thus, the relationship between IL6 and sarcopenia needs further clarify.
Thank you for your suggestion. We modified the text in the “Discussion paragraph” also focusing on the relationship between IL-6 and sarcopenia, and clarifying this concept.
“Our study has some limitations: first of all, this is a single-center study with a rela-tively small sample. The lack of a cirrhotic control group to clarify the role of IL-6 in pre-dicting sarcopenia also in that this context, too, alone could also be considered a limita-tion. Moreover, the lack of a control group consisting of non-cirrhotic HCC patients could be also considered a limitation. IL-6 is a cytokine released during inflammatory processes: it is involved in innate and adaptive immune responses, but also in the activation of met-abolic and catabolic pathways in muscle tissue, depending on the patients condition and muscle state [31]. When there is a prolonged exposure to IL-6, muscle catabolism and at-rophy is observed, probably due a lower level of muscle anabolism and activation of dif-ferent energy production pathways [31]. Hence, IL-6 could be considered a clinical biomarker of sarcopenia and general frailty in patients. In their review, Picca et al. highlighted how, in sarcopenic and frail sub-jects, there was a chronic state of low-grade inflammation, sustained by various stimuli (i.e. gut microbiota, pathogens, cellular debris, or misplaced intracellular components). As a con-sequence, several cytokines (and especially IL-1, IL-6, and TNF-α) exhibited an important role in this context. In particular, IL-6 was associated with frailty and sarcopenia in people aged 75 years [32].
Future studies should focus also on non-HCC cirrhotic patients to confirm our obser-vations and to compare the usefullness of IL-6 in both HCC and non-HCC cirrhotic patients. Nevertheless, IL-6 appears as an effective biomarker for the diagnosis of advanced HCC, particularly in CP score A patients. IL-6n addition, as demonstrated by the logistic regression and multicollinearity model, IL-6 could be considered as a marker of cirrhotic HCC-related sarcopenia when a BIA o CT dedicated software are not available.”
- The etiologies of HCC should be defined in more detail. For example, the definition of alcoholic and metabolic HCC should be described in more detail.
Thank you for the suggestion. We described with more details how the cirrhosis etiologies were defined (line 100-106).
“The inclusion criteria were: signature of the informed consent for data collection, age over eighteen, previous or recent diagnosis of liver cirrhosis including different etiologies (HCV, HBV, alcoholic, metabolic, autoimmune or others) made through liver biopsy, blood tests and imaging, and presence of HCC according to BCLC 2022 criteria not previ-ously treated with systemic therapy [3]. The different etiologies were defined considering the main specific European Association for the Study of the Liver (EASL) guidelines. In particular, we considered as alcohol-related chronic liver disease a daily alcohol con-sumption of 30 g/day, or a weekly consumption 7 drink units in women and 14 drink units in men. [13]. Metabolic chronic liver disease was defined by a biopsy-proven non-alcoholic steatohepatits or with the exclusion of both secondary causes and alcohol con-sumption criteria [14-16]. The exclusion criteria were: heart failure, grade III renal failure and pregnancy.”
- In the section 3.2 Sarcopenia analysis, the results of multivariate linear regression should be provided as a table.
Thanks for the suggestion; we added Table 4 and 5.
- The evaluation of the diagnostic performanceof IL6 showed that AUC values were poor (0.7). It maybe not a good diagnostic biomarker for HCC related sarcopenia
Thank you for the comment. The diagnostic performance of IL-6 could reflect the sample size and the assay sensitivity of 7.8 pg/ml (without discrimination of lower values), but from our point of view this could still be a valid test with a moderate performance, in particular in the absence of a BIA or a CT dedicated software, to detect sarcopenia.
Reviewer 3 Report
The authors analyzed the correlation between IL-6 level, CP score, and sarcopenia in patients with HCC and concluded that IL-6 level is considered as a marker of HCC stages and HCC-related sarcopenia. I’m afraid that this manuscript lacks an important information and a novelty.
1. Choi K, et al. (Clin Mol Hepatol. 2020) previously reported that in patients with HCC serum levels of myostatin, follistatin and IL-6 correlated with sarcopenia. Thus, I concern that there is less novelty in this manuscript.
2. This manuscript lacks important Tables.
The authors should present the results of univariate and multivariate analysis as Tables regarding IL-6, CP, and the severity of HCC. In addition, the association of SMI and IL-6, female sex, BMI, lymphocytes, HCC stage, and CP score should be presented.
3. I cannot evaluate Figure 1 shows that patients with early -stage HCC had a lower CP score and a reduced IL-6. In addition, I cannot also evaluate Figure 2 shows that sarcopenia patients tend toward higher CP score and IL-6 value. If so, Spearman’s rank correlation should be presented.
Author Response
- Choi K, et al. (Clin Mol Hepatol. 2020) previously reported that in patients with HCC serum levels of myostatin, follistatin and IL-6 correlated with sarcopenia. Thus, I concern that there is less novelty in this manuscript.
Thank you for your comment. In agreement with the study conducted by Choi et al. we confirmed the correlation between sarcopenia and the level of IL-6. Moreover we added more information, a ROC curve and a IL-6 cut off were reported, proposing IL-6 as a new marker of sarcopenia.
- This manuscript lacks important Tables. The authors should present the results of univariate and multivariate analysis as Tables regarding IL-6, CP, and the severity of HCC. In addition, the association of SMI and IL-6, female sex, BMI, lymphocytes, HCC stage, and CP score should be presented.
Thanks for the suggestion, we have added Table 3 and Table 4 and 5.
- I cannot evaluate Figure 1 shows that patients with early -stage HCC had a lower CP score and a reduced IL-6. In addition, I cannot also evaluate Figure 2 shows that sarcopenia patients tend toward higher CP score and IL-6 value. If so, Spearman’s rank correlation should be presented
Thanks for the comment. We added in the test the Spearmans’s Test (line 279): “Using Spearman’s test, IL6 and Sarcopenia had p value= 0.014; CP and sarcopenia p was 0.057, while IL6 and CP was 0.007.”
Round 2
Reviewer 1 Report
The authors have improved significantly the manuscript; however, the paper still contains several inadvertences
1. Summary: The correlation between Child Pugh class (CP), Hepatocellular carcinoma (HCC) stage and sarcopenia is still not clear.
Comment: It is not clear why the authors discuss about correlation between sarcopenia and Child Pugh class (CP) since they try to proof that IL 6 is a diagnostic biomarker sarcopenia in advanced–HCC cirrhotic patients.
2. Fig 1 and Fig 2. Contour plot between IL 6, Child Pugh score and HCC stages.
Comment: In the figure appears on the X axis (Child Pugh) value 4 and a part of the curves start from 4 or 4.5. It is not clear why this value appears since Child Pugh score starts with 5.
3. IL 6 can be considered a marker of tumor stage. However this affirmation is not fully supported since IL6 is variable with Child Pugh and Tumor stage ( see multivariate analysis). As can be seen the BCLC C group contains 47 % Child Pugh B and C, which might be one of the reasons for the high values of IL6.
4. IL-6 could can be considered a biomarker for HCC stages and progression. IL 6 is correlated with CP as well, thus it can not be considered a biomarker for HCC progressin.
5. IL6 cold be biomarker for sarcopenia in advanced HCC in cases when CT is not available. The results of the current study have not been validated to support this affirmation
Reviewer 2 Report
This revised manuscript is much improved and all previous comments were responded on point-to-point basis. I have no additional comments.
Reviewer 3 Report
If the authors emphasized that investigation of ROC curve and IL-6 cut off was novel, ROC curve should be presented.
In addition, I cannot evaluate that AUC of 0.72, sensitivity 0.72, and specificity 0.57 are a useful marker of cirrhotic HCC-related sarcopenia.